# Exploring Disparities in Maternal Residential Proximity to Unconventional Gas Development in the Barnett Shale in North Texas

**DOI:** 10.3390/ijerph16030298

**Published:** 2019-01-23

**Authors:** Jennifer Ish, Elaine Symanski, Kristina W. Whitworth

**Affiliations:** 1Department of Epidemiology, Human Genetics & Environmental Sciences, University of Texas Health Science Center at Houston (UTHealth) School of Public Health in San Antonio, San Antonio, TX 78229, USA; Jennifer.L.Epperson@uth.tmc.edu; 2Southwest Center for Occupational and Environmental Health (SWCOEH), Houston, TX 77030, USA; Elaine.Symanski@uth.tmc.edu; 3Department of Epidemiology, Human Genetics & Environmental Sciences, UTHealth School of Public Health, Houston, TX 77030, USA

**Keywords:** unconventional gas development, index of concentration at the extremes, maternal and child health, Barnett Shale, environmental justice

## Abstract

*Background*: This study explores sociodemographic disparities in residential proximity to unconventional gas development (UGD) among pregnant women. *Methods*: We conducted a secondary analysis using data from a retrospective birth cohort of 164,658 women with a live birth or fetal death from November 2010 to 2012 in the 24-county area comprising the Barnett Shale play, in North Texas. We considered both individual- and census tract-level indicators of sociodemographic status and computed Indexes of Concentration at the Extremes (ICE) to quantify relative neighborhood-level privilege/disadvantage. We used negative binomial regression to investigate the relation between these variables and the count of active UGD wells within 0.8 km of the home during gestation. We calculated count ratios (CR) and 95% confidence intervals (CI) to describe associations. *Results*: There were fewer wells located near homes of women of color living in low-income areas compared to non-Hispanic white women living in more privileged neighborhoods (ICE race/ethnicity + income: CR = 0.51, 95% CI = 0.48–0.55). *Conclusions*: While these results highlight a potential disparity in residential proximity to UGD in the Barnett Shale, they do not provide evidence of an environmental justice (EJ) issue nor negate findings of environmental injustice in other regions.

## 1. Introduction

Persistent socioeconomic and racial disparities in birth outcomes are not fully understood. Inequalities in the distribution of environmental hazards also exist, with low-income communities and communities of color bearing a greater burden of exposure to environmental contaminants [1], many of which have been associated with adverse birth outcomes [2]. Thus, environmental factors may play a role in observed perinatal health disparities. These disparities may also be explained in part by the combined effects of individual- and community-level psychosocial stressors which increase allostatic load [3], resulting in a heightened susceptibility to environmental hazards and increasing women’s risk of adverse birth outcomes [4]. 

Such “double jeopardy” of chemical and non-chemical stressors [5] (p. 6) may exist in residential areas near unconventional gas development (UGD), i.e., horizontal drilling and hydraulic fracturing. UGD has dramatically increased over the past two decades and has expanded into more densely populated areas [6], raising public health concerns for nearby residents [7]. In addition to potential air pollution and groundwater contamination, UGD is a source of noise and light pollution, unpleasant odors, and increased psychosocial stressors [7,8,9,10]. These factors may contribute to adverse perinatal health outcomes, such as preterm birth, low birth weight, and possibly certain birth defects, that have been associated with living near UGD activity in previous epidemiologic studies [11,12,13,14].

The increase in psychosocial stressors [7,8] associated with living near UGD may have a disproportionate impact on already disadvantaged populations, potentially exacerbating existing health disparities. A limited body of literature raises concerns that UGD wells are disproportionately located near disadvantaged neighborhoods compared to more advantaged neighborhoods, although evidence is mixed [15]. In areas of Pennsylvania overlaying the Marcellus Shale, Ogneva-Himmelberger and Huang [16] found a higher proportion of people who live below the poverty line in census tracts with a high versus low UGD well density. Also in Pennsylvania, Clough and Bell [17] found neither race-based nor poverty-based disparities when examining the sociodemographic composition of communities near UGD. However, in Colorado, McKenzie et al. [18] found economic-based disparities in areas with substantial UGD, observing an increase in the proportion of low-value homes within one mile of a UGD well from 2000 to 2012. In the Eagle Ford Shale in South Texas, Johnston et al. [19] observed that 26.4% of residents of color lived within 5 kilometers of a well compared to 29.8% of non-Hispanic white residents.

The issue of whether any communities are disproportionately affected by UGD has not been fully explored. To date, existing UGD-related environmental justice (EJ) investigations have primarily been conducted in rural areas and/or areas with low racial diversity [16,17,18,19]. Furthermore, there is a lack of individual-level demographic data in previous studies. Except for one study by McKenzie et al. [18], all analyses have been conducted primarily at the census-tract level.

The Barnett Shale in North Texas is home to urban and suburban communities and a racially diverse population; it has also seen substantial UGD in populated areas [20]. This is in contrast to UGD in other areas of the country (e.g., in the Marcellus Shale in Pennsylvania) and in Texas (e.g., the Eagle Ford Shale in South Texas) where UGD occurs primarily in rural areas. An investigation of racial and economic disparities in proximity to UGD in the Barnett Shale provides a more complete understanding of the distribution of potential environmental exposures from UGD operations, as evidence from other studies may not be generalizable to communities near the Barnett Shale. Thus, the purpose of this study was to determine whether there are sociodemographic disparities in residential proximity to UGD among pregnant women in communities surrounding the Barnett Shale. A more complete understanding of disparities in environmental health risks associated with UGD could inform potential mitigation strategies and thus improvements in maternal and child health outcomes.

## 2. Materials and Methods

We conducted a secondary data analysis based on a retrospective birth cohort among 164,658 women living in the 24-county Barnett Shale area with a birth (*n* = 163,827) or fetal death (*n* = 831) from 30 November 2010 to 29 November 2012. Details regarding the parent study have been previously published but are briefly discussed here [14]. The cohort was identified based on birth and fetal death records from the Texas Department of State Health Services (TXDSHS) from which we also obtained maternal age (≤20, 21–25, 26–30, 31–35, ≥36 years), education (<high school, high school graduate, some college, college graduate), and race/ethnicity (non-Hispanic white, non-Hispanic black, Hispanic, other). The Adequacy of Prenatal Care Utilization Index (inadequate, intermediate, adequate, adequate plus, unknown) developed by Kotelchuck [21] was calculated using the reported timing of the first prenatal visit and frequency of visits from the birth or fetal death record. In this study, we considered the Adequacy of Prenatal Care Utilization Index as a proxy of access to health care and socioeconomic status. Maternal address at delivery was also obtained from TXDSHS records and geocoded to the street level.

In the original study, the locations of 14,351 unconventional (i.e., horizontally or directionally drilled) gas wells in the Barnett Shale that were active between 1 January 2010 and 29 November 2012 were obtained from DrillingInfo (www.drillinginfo.com) and linked to geocoded maternal address at delivery. For the present analysis, the dependent variable was defined as the total number of active UGD wells ≤0.8 km of the maternal residence during the gestational period, modeled in continuous form.

We obtained data from the 2010 American Community Survey (ACS) 5-year estimates for each of the 1,218 census tracts in the 24-county study area [22]. The women in this study resided in 812 of these census tracts. We used census-tract level data for educational attainment, race/ethnicity, and household income, chosen based on their implication in previous studies of UGD-related environmental justice [16,17,19]. We also obtained data on %unemployed, %female-headed households, and %crowded housing for each census tract. All census tract-level covariates were categorized into quartiles for analysis. Lastly, we classified each census tract as urban or rural based on the 2010 US Census estimate of the percentage of the population in the respective county living in rural areas [23]. Dallas, Denton, and Tarrant counties had a median rural population of 1.29% (range = 0.69%–6.91%) and were classified as urban. The other 21 counties had a median rural population of 58.94% (range = 19.42%–100%) and were classified as rural.

We computed census tract-level Index of Concentration at the Extremes (ICE) metrics for education and for race/ethnicity and income combined. These metrics are based on the work of Massey [24] and measure relative deprivation and privilege in a specified geographical area. Unlike other measures of residential segregation (e.g., the Gini index [25]), the ICE considers the number of affluent persons or households, thereby quantifying the “proportional imbalance between affluence and poverty” rather than relying on separate indices [24] (p. 46). Krieger et al. [26] further developed this construct by creating novel ICE metrics to examine racial disparities in exposure to environmental contaminants. The general formula for the ICE is:ICE = (A_i_ − P_i_)/T_i_(1)
where A_i_ represents the number of units (e.g., individuals or households) in the privileged group, P_i_ represents the number of units in the least privileged group, and T_i_ is the total number of units in the census tract. Following Krieger et al. [26], we calculated an ICE metric for race/ethnicity and income, where A_i_ = number of households in the census tract that are non-Hispanic white and earning at least $100,000, P_i_ = number of households in the census tract that are non-white and earning less than $25,000 annually and T_i_ = total number of households in the census tract. The use of a single index for race/ethnicity and income is recommended over separate indices because it more meaningfully describes the combined impact of economic and racial/ethnic segregation [27]. We also calculated an ICE metric for education, where the most and least privileged groups were set as the number of individuals having earned at least an Associate’s degree and individuals earning less than a high school diploma, respectively. The ICE can range from −1 to 1, with a value of 1 indicating that all residents or households are in the most privileged group (i.e., concentrated privilege), and a value of −1 indicating that all residents or households are in the least privileged group (i.e., concentrated disadvantage). Women in our study were assigned ICE values based on the census tract of residence reported on the birth or fetal death record.

We conducted descriptive analyses to summarize study population characteristics with respect to each individual- and census tract-level covariate. We examined each covariate in both univariable and multivariable negative binomial regression models. Count ratios (CR) and 95% confidence intervals (CI) were calculated to describe the association between each indivdiual- and census tract-level covariate and maternal residential proximity to UGD. Given concerns that the determinants of residential proximity to UGD may differ between rural and urban areas, we stratified the final multivariable analysis by rurality. All analyses were performed using SAS version 9.4 (SAS Institute Inc., Cary, NC, USA).

The study protocol was approved by the UTHealth Committee for the Protection of Human Subjects (HSC-SPH-13-0151) and by the TXDSHS IRB (13-037).

## 3. Results

Among the cohort, 15.3% of women had at least one active UGD well within 0.8 km (1/2 mile) of her home during pregnancy. The median number of wells among these women was three, and the greatest number of wells was 32 (data not shown). The largest racial/ethnic group was Hispanic (39.7%), followed by non-Hispanic white (37.5%) and non-Hispanic black (16.0%) (Table 1). Most of the women were younger than 31 years of age (68.7%) and had at least a high school education (79.2%). Just over 40% had adequate prenatal care, and 36.2% of the women had less than adequate prenatal care. Most women (87.3%) lived in the three most urban counties in the study area (i.e., Dallas, Denton, and Tarrant). The values for ICE race/ethnicity and income among the study population ranged from −0.87 to 1 (median = −0.01, IQR = −0.18–0.15). Similarly, for ICE education, values ranged from −0.81 to 1 (median = 0.12, IQR = −0.17–0.33), indicating a greater number of women in census tracts that favor higher educational attainment.

In univariable analyses, compared to the most privileged women (i.e., the highest quartile of ICE race/ethnicity and income), women classified in any lower quartile had fewer active wells near their homes (Table 2). Similarly, women who were Hispanic or non-Hispanic black lived near fewer UGD wells than white women. Women living in census tracts with lower educational attainment (i.e., the first and second quartile of ICE education) had slightly fewer UGD wells near their homes than women who lived in the most highly educated neighborhood. In addition, women living in census tracts in the highest quartile of unemployment rates or %crowded housing also had fewer active UGD wells near their residences.

Due to the high correlation between each of the census tract-level variables and ICE metrics (data not shown; Spearman’s *r* ranged from 0.41 to 0.77) and the strong univariable association of the ICE for race/ethnicity and income with UGD activity, this ICE metric was the only indicator of census tract characteristics included in the multivariable model (Table 2). Results from this model were similar to the univariable results. Relative to women living in neighborhoods in the highest quartile of the combined race/ethnicity and income ICE index (i.e., the most privileged), women living in neighborhoods in the third (CR = 0.70, 95% CI = 0.66–0.74), second (CR = 0.62, 95% CI = 0.58–0.66), and lowest (CR = 0.49, 95% CI = 0.46–0.53) quartiles of the ICE for race/ethnicity and income had fewer active wells near their homes. Compared to non-Hispanic white women, fewer wells were located near homes of women of color. These associations were slightly weaker in the multivariable analysis compared to the univariable analysis but were similar for non-Hispanic black women (CR = 0.87, 95% CI = 0.82–0.92) and Hispanic women (CR = 0.71, 95% CI = 0.68–0.75). Contrary to the univariable results, in the multivariable model, college-educated women had fewer active UGD wells near their homes than less educated women. On average, a slightly higher number of UGD wells were located near mothers in their twenties compared to mothers 20 years-old or younger, and women who had inadequate prenatal care had a greater number of UGD wells near her home than women who received adequate care. To investigate potential differences in the associations of sociodemographic factors and UGD activity in rural and urban areas, we also stratified the multivariable model by rurality. In these models, we categorized the ICE for race/ethnicity and income into tertiles rather than quaritles due to small numbers in rural counties. We observed no meaningful differences in associations of sociodemographic covariates and residental proximity to UGD between urban and rural counties (Appendix A).

## 4. Discussion

In our study, we found more UGD activity near the homes of racially and economically privileged women compared to disadvantaged women. Women of color had fewer UGD wells near their homes during pregnancy compared to non-Hispanic white women. Similarly, women living in the most socioeconomically disadvantaged neighborhoods had fewer wells near their homes relative to women in the most privileged neighborhoods. Thus, in communities surrounding the Barnett Shale, it appears that non-Hispanic white women and affluent neighborhoods are most affected by UGD activity. Our results may be explained, in part, by the fact that most UGD drilling activity in the Barnett Shale occurs near suburbs of Fort Worth and west of Tarrant County. In contrast to the increasing racial diversity of Dallas county [28], these areas remain majority non-Hispanic white [22].

The association between adequacy of prenatal care (which we considered a proxy of health care access and socioeconomic status) and UGD activity was inconsistent with results for race/ethnicity and income. Given that the positive association between privilege and proximity to UGD activity was consistent across most other individual- and census tract-level covariates, this may indicate that prenatal care is a poorer proxy of socioeconomic status than race/ethnicity and income in this population. It could also be that prenatal care utilization is a proxy for other factors such as cultural differences or healthcare availability. The finding that less educated women had more UGD wells near their homes was also inconsistent with our main results, but the reasons for this association are unclear.

Results from existing EJ literature regarding UGD are mixed [15]. In line with our study, Clough and Bell [17] found no evidence that low-income or minority communities were disproportionately represented in communities near UGD in the Marcellus Shale. Similarly, Johnston et al. [19] report a lower prevalence of people of color living within 5 kilometers of an unconventional oil or gas well compared to non-Hispanic white residents in South Texas (26.4% vs. 29.8%, respectively). The researchers suggest these patterns might be explained by the higher proportion of non-Hispanic white landowners in the Eagle Ford Shale, meaning that this group also likely receives economic benefits from unconventional oil and gas development operations that occur on privately owned property. Given the differences in urbanicity and land-use patterns between the Eagle Ford Shale area and the Barnett Shale area, it is unclear if a similar explanation holds true in our population. Our overall findings are inconsistent with two previous investigations of UGD that provide evidence that low-income or minority communities are disproportionately affected by UGD in Pennsylvania and Colorado [16,18]. It is important to note that, expect for one study by McKenzie et al. [18], previous studies relied on group-level data rather than individual-level data. When we explored census tract-level indicators of race or income as used in previous studies (e.g., %minority, %low-income), the results corroborated the findings for the ICE metric.

Equivocal results across previous studies may be reflected in the different study populations and methods used to determine the sociodemographic characteristics of those populations. Existing UGD-related EJ studies have been conducted in rural regions in Pennsylvania and South Texas, while this study was the first EJ investigation of UGD in a mainly urban and suburban region in North Texas with a racially and ethnically diverse population. There may also be other differences in population demographics, land use, and policies regulating UGD that provide different contexts for UGD operations between the Barnett Shale counties and regions studied in previous investigations and within Barnett Shale counties themselves. For example, part of the legal framework in Texas that governs UGD is a patchwork that varies by municipality, demonstrated by a case study of environmental justice concerns over UGD in Denton, TX [29].

Our study is the first to utilize an individual-level metric of residential proximity to UGD as well as individual-level sociodemographic characteristics to examine disparities among women in a diverse Texas region. This includes the use of ICE metrics to characterize neighborhood characteristics for each woman in the study population. As is the case in most retrospective birth cohorts, a limitation of our study was incomplete information on maternal residential history. Because the number of UGD wells near a woman’s home was assigned based on maternal residence at birth, maternal residential mobility during pregnancy could have led to misclassification in maternal residential proximity to UGD. Given that lower socioeconomic status is associated with higher maternal residential mobility, it is possible that bias was introduced in our effect estimates [30]. Additionally, we were not able to investigate income at the individual-level. Furthermore, a full understanding of factors that determine UGD well locations would require consideration of geographical, regulatory, and political factors (e.g., land suitability, feasibility of transportation to the well site, land and mineral rights ownership, etc.), and a complete understanding of residential proximity to UGD wells would require knowledge of individual behaviors regarding where to live.

## 5. Conclusions

We found that non-Hispanic white women and women living in more privileged neighborhoods had greater numbers of active UGD wells in close proximity to their residence during their pregnancy, as compared with women of color and those living in less privileged areas. While these results highlight a potential disparity in residential proximity to UGD wells, it does not provide evidence of an environmental justice issue in the Barnett Shale as related to UGD activity. Importantly, our findings may not be generalizable to other regions due to differences in the landscape of UGD across the United States.

## Figures and Tables

**Table 1 ijerph-16-00298-t001:** Descriptive characteristics of 164,658 women with a singleton birth or fetal death in the 24-county Barnett Shale area, Texas, 2010–2012.

	Total Cohort (*N* = 164,658)
Maternal age (years), %	
≤20	15.1
21–25	25.1
26–30	28.5
31–35	21.2
36+	10.1
Maternal education, %	
College graduate	23.8
Some college	24.8
High school graduate	30.6
Less than high school	20.8
Missing	<0.1
Maternal race/ethnicity, %	
Non-Hispanic White	37.5
Hispanic	39.7
Non-Hispanic Black	16.0
Other	6.7
Adequacy of prenatal care, %	
Inadequate	21.4
Intermediate	14.8
Adequate	41.2
Adequate plus	18.2
Unknown	4.5
Rurality ^1^, %	
Urban	87.3
Rural	12.7
ICE Race/ethnicity and income, median (IQR) ^2^	−0.01 (−0.18, 0.15)
ICE Education, median (IQR) ^2^	0.12 (−0.17, 0.33)
% Unemployed, median (IQR) ^2^	6.3 (4.2, 9.2)
% Female-headed households, median (IQR) ^2^	14.2 (9.3, 21.6)
% Crowded housing, median (IQR) ^2^	3.6 (1.3, 8.0)

IQR = Interquartile Range; ICE = Index of Concentration at the Extremes (−1 = extreme concentration of deprivation; 1 = extreme concentration of privilege); ^1^ Rurality of county of residence; ^2^
*n* = 3 missing.

**Table 2 ijerph-16-00298-t002:** Associations between sociodemographic variables and the count of active unconventional gas development (UGD) wells within 0.8 km of residential address among 164,658 women with a singleton birth in the 24-county Barnett Shale area, Texas, 2010–2012.

	Univariable			Multivariable ^1^	
	Count Ratio	95% CI		Count Ratio	95% CI
*Individual-Level Variables*					
Maternal age (years)					
≤20	1.0 (ref)			1.0 (ref)	
21–25	1.10	(1.03, 1.17)		1.09	(1.02, 1.16)
26–30	1.15	(1.08, 1.22)		1.12	(1.05, 1.19)
31–35	1.06	(1.00, 1.13)		1.04	(0.96, 1.10)
36+	1.02	(0.94, 1.11)		1.02	(0.93, 1.09)
Maternal education					
College graduate	1.0 (ref)			1.0 (ref)	
Some college	1.16	(1.04, 1.22)		1.34	(1.26, 1.41)
High school graduate	0.91	(0.86, 0.96)		1.20	(1.13, 1.27)
Less than high school	0.96	(0.91, 1.02)		1.48	(1.38, 1.58)
Maternal race/ethnicity					
Non-Hispanic White	1.0 (ref)			1.0 (ref)	
Hispanic	0.64	(0.61, 0.66)		0.71	(0.68, 0.75)
Non-Hispanic Black	0.72	(0.68, 0.77)		0.87	(0.82, 0.92)
Other	0.69	(0.64, 0.75)		0.68	(0.62, 0.75)
Adequacy of prenatal care					
Adequate	1.0 (ref)			1.0 (ref)	
Inadequate	1.08	(1.03, 1.14)		1.19	(1.13, 1.26)
Intermediate	1.02	(0.69, 1.08)		1.01	(0.95, 1.07)
Adequate plus	0.87	(0.83, 0.92)		0.86	(0.81, 0.91)
Unknown	0.69	(0.63, 0.76)		0.68	(0.62, 0.75)
*Census Tract-Level Variables*					
ICE Race/ethnicity and income ^2^					
Q4 (high)	1.0 (ref)			1.0 (ref)	
Q3	0.71	(0.67, 0.75)		0.70	(0.66, 0.74)
Q2	0.62	(0.59, 0.65)		0.62	(0.58, 0.66)
Q1 (low)	0.49	(0.47, 0.52)		0.49	(0.46, 0.53)
ICE Education ^3^					
Q4 (high)	1.0 (ref)			-	
Q3	1.39	(1.32, 1.47)		-	
Q2	0.91	(0.86, 0.96)		-	
Q1 (low)	0.90	(0.85, 0.95)		-	
%Unemployed ^4^					
Q1	1.0 (ref)			-	
Q2	0.90	(0.85, 0.95)		-	
Q3	1.06	(1.01, 1.12)		-	
Q4	0.89	(0.85, 0.94)		-	
%Female-headed households ^5^					
Q1	1.0 (ref)			-	
Q2	1.02	(0.97, 1.08)		-	
Q3	0.67	(0.63, 0.70)		-	
Q4	0.80	(0.76, 0.84)		-	
%Crowded housing ^6^					
Q1	1.0 (ref)			-	
Q2	1.05	(1.00, 1.11)		-	
Q3	0.92	(0.87, 0.97)		-	
Q4	0.57	(0.54, 0.61)		-	

95% CI = 95% confidence interval; ICE = Index of Concentration at the Extremes; Q = quartile; ^1^ Included maternal age (years), maternal education, maternal race/ethnicity, prenatal care utilization index, and ICE race/ethnicity and income; ^2^ ICE race/ethnicity and income Q1: −0.874 to <−0.183; Q2: −0.183 to <−0.011; Q3: −0.011 to <0.150; Q4: 0.150 to 1; ^3^ ICE education Q1: −0.811 to <−0.168, Q2: −0.168 to <0.122, Q3: 0.122 to <0.330, Q4: 0.330 to 1; ^4^ Unemployment rate Q1: 0.0 to <4.2, Q2: 4.2 to <6.3, Q3: 6.3 to <9.2, Q4: 9.2 to ≤36.6; ^5^ Female-headed households Q1: 0.0 to <9.3, Q2: 9.3 to <14.2, Q3: 14.2 to <21.6, Q4: 21.6 to ≤66.7; ^6^ Crowded housing Q1: 0.0 to <1.3, Q2: 1.3 to <3.6, Q3: 3.6 to <8.0, Q4: 8.0 to ≤39.8.

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
