# Peer review of "Exploring Disparities in Maternal Residential Proximity to Unconventional Gas Development in the Barnett Shale in North Texas"

_ijerph, 2019, doi:10.3390/ijerph16030298_

Round 1
Reviewer 1 Report
Line 148: insert space between “to” and the numeral “1”
Lines 158-159: there is a word missing in this sentence.
The results text should be revised to more precisely reflect the numerical results. I observe several inconsistencies. For example:
Line 181: this statement does not match the results presented in Table 2. In both univariable and multivariable models, women with inadequate care had a higher well count compared to those with adequate care, and women with more than adequate care had fewer wells near their homes.
Line: In univariable models, “tracts with higher unemployment had fewer active UGD wells near their residences.” This statement is not quite accurate, as the relationship is not monotonic across the quartiles. It oversimplifies what was observed.
It would be helpful if Table 2 clearly showed which variables are individual level and which are census tract level.
Line 218: I would disagree with the statement that this study is inconsistent with the existing literature, which I find to interpret to have mixed findings. This study is generally consistent with prior studies, which do not point to a disparity in the expected direction when it comes to well placement. This may have to do with land ownership, rurality/suburban/ whiteness. Tying this into the discussion about these point would be useful.
Reviewer 2 Report
Major comments:
1. The authors explore the relationship between a number of population characteristics among pregnant women in the Barnett shale region and the count of nearby gas wells. To explore the relationship, they employ creative use of Census-tract and individual-level data along with a Census tract-level inequality index. The main strength of the study is its use of individual-level characteristics.
2. Combining data for women living in Census tracts diminishes the use of the individual-level data, especially with individual-level well count data. It is also not clear how the authors managed any correlation between individual-level race and ethnicity and census-tract level race and ethnicity. Combining individual-level and Census tract-level variables confuses interpretation of the multivariable model. What value is added by combining individual-level and Census tract-level variables into a single model?
3. The authors report that “[de]tails regarding this parent study have been previously published but are briefly discussed here,” citing a 2017 epidemiology study “ Maternal Residential Proximity to Unconventional Gas Development and Perinatal Outcomes among a Diverse Urban Population in Texas. It would be worthwhile for the authors It to describe how this secondary analysis influences interpretation of results from their earlier work, if at all.
Other comments:
1. It would be useful to explain very briefly how the authors ensured that their well counts include only UGD wells and not conventional wells.
2. Was there a correlation between
3. Racial/income inequality can be higher in non-rural areas, so it would be useful to report results from the models stratified by rurality.
4. Could prenatal care utilization be a proxy for other factors, such as cultural differences or healthcare availability in areas with dense UGD?
Minor comments
1. Why did the authors not use census-block data, which have higher temporal resolution than census-track data?
2. Page 2, line 47: Suggest providing citations to support “increase in psychosocial stressors associated with living near UGD…”
3. There are quite a few typographical errors in the manuscript that require correction.
